# The cost and health-related quality of life of stroke management and care of acutely hospitalized cases in Mozambique

Igor Samuel Dobe◉[1,2,3]*, Clifford Afoakwah◉[4,5◉], Neide Canana[2◉], Simon Stewart[1,6‡], Ana Mocumbi[1,2‡]

**1** Universidade Eduardo Mondlane, Maputo, Moçambique, **2** Instituto Nacional de Saúde, Maputo, Moçambique, **3** Hospital Geral de Mavalane, Maputo, Moçambique, **4** Australian Centre for Health Services Innovation, School of Public Health and Social Work, Queensland University of Technology, Kelvin Grove, Australia, **5** Jamieson Trauma Institute, Metro North Health, Herston, Australia, **6** University of Notre Dame Australia, Fremantle, Western Australia

◉ These authors contributed equally to this work.
‡ SS and AM also contributed equally to this work.
* igor.dobe@ins.gov.mz

## Abstract

### Background

Stroke is a leading cause of death and disability, placing a significant burden on survivors and their families. To address the lack of African-specific data, we investigated the cost of stroke management and the recovery of health-related quality of life in the post-discharge period in Mozambique.

### Methods

A prospective, cost-of-illness study examining the direct and indirect costs of acute stroke presentations to a first referral urban public hospital in Maputo, Mozambique (June-December 2019). Direct costs were derived from medical records to estimate the cost of hospital care. Indirect costs were derived from interviews using a semi-structured questionnaire administered to patients or their caregivers during the index hospitalization and 28-days post-discharge to estimate additional expenditure and loss of productivity due to disability which varied by employment status (informal, formal, pensioner and unemployed). Health-related quality-of-life was assessed at 28-days post stroke using the EQ-5D-3L questionnaire. Cost analysis was conducted from a societal perspective and reported in $USD.

### Results

50 of 80 patients admitted with an acute stroke were consecutively recruited during the study period. Median age was 61 (IQR 38–68) years, 56% were women and 44% presented with a hemorrhagic stroke. Median length of stay in the hospital was 7.0

**Data availability statement:** All data are available from the figshare database: https://figshare.com/s/e0ac4ee66c38bf1af930 DOI: 10.6084/m9.figshare.26505883

**Funding:** The author(s) received no specific funding for this work.

**Competing interests:** The authors have declared that no competing interests exist.

(IQR 4.0 to 8.0) days. Within 28-days post-discharge 20% patients had died. Estimated total direct cost of hospital care for 50 patients (hospital days, medication, and investigations) was $36,315.28, the median cost per patient was $721.45 (IQR 582; $790). Estimated direct non-medical costs per patient during hospitalization median $12,59 (IQR, 8.19; 16.39) and mean $13.62 (SD 8.02). In the first 28 days after discharge the non-medical cost was: $32.04 (IQR, 19,01; 49.83) and mean $41.37 (SD, 36.11). Overall, loss of productivity was very high in informal employment and quality of life in survivors severely compromised. The mean EQ-5D index and VAS scores of stroke patients were 0.514 (SD, 0.298), and 49.39 (SD, 20.95), respectively. Anxiety/depression 92.5% and Pain/discomfort 82.5% were the most frequently reported issues.

## Conclusion

The economic cost of stroke in low-income sub-Saharan African countries such as Mozambique is substantially high, with considerable out-of-pocket spending, poor survival rate and a compromised health-related quality-of-life. Health system reforms designed to mitigate the individual to societal burden imposed by stroke are required.

## Introduction

Stroke ranks as the second leading cause of death worldwide, responsible for approximately 6.6 million deaths in 2021, according to the most recent Global Burden of Disease estimates [1]. It is now recognized as a major cause of death in sub-Saharan Africa as well [2]. For survivors, stroke often leads to persistent disabilities related to motor and cognitive function, significantly impairing quality of life [3].

In Mozambique, stroke represents a growing burden associated with the ongoing epidemiological transition. According to national and global estimates, stroke has become one of the top ten causes of death in the country, with a 25% increase in stroke-related mortality between 2009 and 2019 [4,5]. Although by 2019 stroke ranked as the second leading cause of death in Mozambique [6], historical data from 2009 are insufficiently detailed to determine its exact position in the ranking at that time. Therefore, while we emphasize this rising trend, we acknowledge the limitations in earlier data sources.

Low standards of care, limited access to health services and medications, and delays in hospital presentation contribute to the heavy impact of stroke in the region [7]. Despite the importance of understanding stroke's cost dynamics in this setting, there is a lack of studies examining the economic burden of stroke care in Africa. This gap limits the ability to inform health service planning and research priorities related to the prevention and treatment of stroke.

In Mozambique, poverty, low health literacy, and limited health system capacity significantly hinder the implementation of effective stroke prevention and management strategies. In this context, estimating both the direct and indirect costs of stroke is essential for evaluating how limited healthcare resources can be optimally

allocated [8]. Such cost data are also fundamental for conducting robust economic evaluations of treatment strategies aimed at improving clinical outcomes for individuals affected by stroke.

Therefore, this study aimed to estimate the economic burden of stroke care during hospitalization and in the early post-discharge period in Mozambique. Additionally, we assessed health-related quality of life (HRQoL) 28 days after hospital discharge to capture the broader societal impact of stroke-related disability. This perspective is particularly relevant in a country where formal rehabilitation and long-term support systems are still limited.

Recent research conducted at Hospital Provincial da Matola in Maputo province has shown that stroke survivors in Mozambique often experience serious physical and functional impairments such as dysphagia that negatively affect their quality of life, even when the perceived severity is low [9]. Moreover, a systematic review of African studies found that stroke survivors consistently report lower HRQoL than the general population, with post-stroke dependence and depression identified as key contributing factors [10]. By incorporating HRQoL outcomes, this study adds a patient-centered dimension that can inform the development of more effective rehabilitation strategies and health policies in resource-limited settings.

## Materials and methods

### Study design

We conducted a cross-sectional, prospective, incidence-based cost-of-illness study from a societal perspective. The study was carried out in a consecutively recruited cohort of adult patients presenting with a first-ever acute stroke at Mavalane General Hospital, a tertiary-referral facility in Maputo, Mozambique. The objective was to estimate both the direct costs to the healthcare system and the indirect costs to patients and their families.

In addition, the study included a health-related quality of life (HRQoL) assessment using the EQ-5D-3L instrument, conducted 28 days after hospital discharge, to evaluate the broader impact of stroke on patients' daily functioning and well-being in a low-resource setting.

### Study cohort

All patients aged 18 years or older admitted to the Internal Medicine ward of Mavalane General Hospital in Maputo, Mozambique, between June and December 2019 were screened for eligibility. Patient identification was performed through daily review of hospital admission registries to minimize selection bias.

Eligible participants were those who had been hospitalized within 72 hours of the onset of a first-ever stroke (ischemic or hemorrhagic), as defined by the World Health Organization [11], and who consented to participate in a home-based follow-up visit 28 days after hospital discharge. Patients residing in remote areas beyond the hospital's follow-up capacity or those unable to complete follow-up at 28 days post-discharge were excluded. Stroke diagnosis and subtype classification were confirmed by the attending clinician based on computed tomography (CT) imaging. The recruitment process and follow-up of stroke cases throughout the study period are summarized in Fig 1.

### Data collection

Data were collected by trained investigators during hospitalization and through in-person home visits conducted 28 days after discharge, using a semi-structured questionnaire.

**Demographics and clinical characteristics.** Sociodemographic and clinical characteristics were collected using a structured questionnaire administered during hospitalization and during follow-up at 28 days post-discharge.

The sociodemographic variables collected included age, sex, ethnicity, marital status, education level, income level, and employment status. Employment status was classified as: Unemployed: defined as individuals who were able and willing to work, were available for employment, and were actively seeking a job prior to the stroke event but were not currently employed.

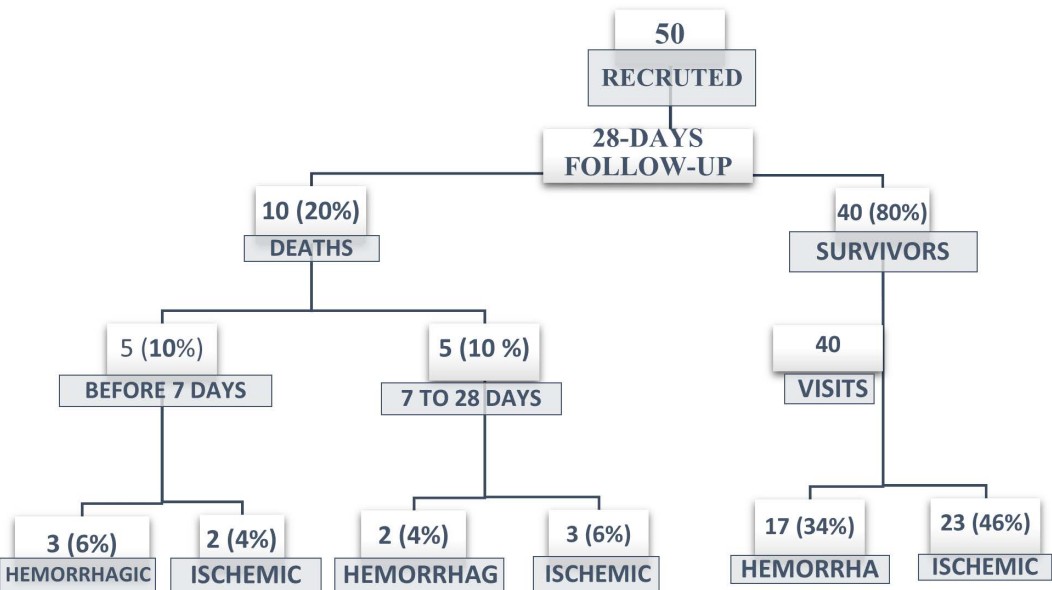

**Fig 1. Flow diagram of participant recruitment and follow-up process.**

Type of employment: categorized as either formal or informal. Informal employment was defined as work that is not regulated by national labor legislation and does not provide access to core social protections, such as paid sick leave, maternity benefits, retirement pensions, or health insurance [12–14]. Formal employment was defined as work regulated under national labor laws, with access to statutory social protection schemes, such as retirement pensions, work injury compensation, and benefits administered through the National Institute of Social Security (INSS) in Mozambique [12,13].

Clinical data included stroke subtype (ischemic or hemorrhagic), confirmed by computed tomography (CT) imaging, length of hospital stay, and presence of stroke risk factors such as hypertension, diabetes mellitus, and HIV infection. Stroke was defined according to the Tenth Revision of the International Classification of Diseases (ICD-10) [15].

**Cost assessment and calculation.** This study followed standard cost-of-illness methodology from a societal perspective, incorporating both direct and indirect costs related to stroke care. The approach was guided by the general principles of cost-of-illness analysis and the recommendations of the World Health Organization's CHOICE (Choosing Interventions that are Cost-Effective) framework [16].

## Cost categories

Costs were categorized as:

(i) **Direct medical costs:** health system resources used per patient, including hospital admission, inpatient stay, diagnostic imaging (e.g., laboratory, radiology, and cardiology tests), pharmacotherapy, and procedures. Unit prices for medications were obtained from the WHO/HAI Median Price Ratio database [17].

(ii) **Direct non-medical costs:** out-of-pocket expenses borne by patients and families, including transport, food, hygiene items, and other non-clinical expenses incurred during hospitalization and the first 28 days post-discharge.

(iii) **Indirect costs:** productivity loss associated with stroke-related disability, calculated using the human capital approach. These included days of work missed by patients and informal caregivers, both during hospitalization and in the post-discharge period [18].

For patients engaged in informal employment, productivity losses were estimated based on self-reported average daily income prior to the stroke. Given the variability in informal sector income, a triangulation method was applied using participant self-reports, local wage benchmarks, and national statistics to generate conservative and contextually relevant estimates [19,20].

**Hospitalization costs.** Data on costs associated with inpatient care were collected from patient medical records and included: Admission fees, Duration of hospital stay, Medications, Diagnostic tests and procedures, In-hospital meals and transport to/from the facility, Caregiver involvement and associated productivity loss.

**Post-discharge costs (28 days).** Post-discharge data were collected through in-person home visits conducted 28 days after hospital discharge. Information was gathered on: Medication and follow-up consultations, Laboratory and imaging costs, Patient and caregiver productivity loss, Transportation to outpatient or rehabilitation services, Home adaptations made as a result of stroke-related disability, Daily expenditures on food, hygiene, and comorbidity management.

Out-of-pocket expenditures captured at 28 days post-discharge included all health-related spending, regardless of whether it was directly attributable to stroke or associated comorbidities. Given the limitations of patient recall and the difficulty in separating stroke-specific from comorbidity-related expenses, reported values were aggregated, which may introduce some confounding.

Out-of-pocket costs collected at 28 days post-discharge reflected all health-related expenditures, including those potentially associated with pre-existing comorbidities. Due to challenges in disaggregating stroke-specific versus comorbidity-related costs through patient recall, these were reported as aggregate values, which may introduce some degree of confounding.

## Assessment of health-related quality of life

Health-related quality of life (HRQoL) was evaluated 28 days after hospital discharge using a multidimensional approach that captures the impact of stroke on physical, emotional, and social functioning. The Portuguese version of the EQ-5D-3L instrument was used to measure HRQoL. This tool assesses five health dimensions mobility, self-care, usual activities, pain/discomfort, and anxiety/depression each with three levels of severity, resulting in 243 possible health states [21]. These states are converted into a single index value ranging from 0 (representing death) to 1 (representing full health) [22].

The EQ-5D-3L was chosen over the more recent 5-level version (EQ-5D-5L) for two key reasons: (1) the lack of a validated Portuguese translation of the EQ-5D-5L at the time of protocol development, and (2) evidence from previous studies indicating challenges in distinguishing between the five response levels in populations with low literacy [22]. These factors made the 3L version more suitable for the study context in Mozambique.

As no country-specific value set exists for Mozambique, index scores were derived using valuation coefficients from a time trade-off (TTO) study conducted in Zimbabwe, a sub-Saharan African country with similar socioeconomic conditions and linguistic heritage [23]. While not ideal, this approach has been previously used in other low-resource settings in the absence of national value sets and provides a contextually relevant approximation of utility weights.

## Caregiver time and cost estimation

Informal caregiving needs were assessed during the same 28-day post-discharge period. Caregivers were defined as unpaid individuals, typically family members or close friends, who assisted patients with daily activities such as mobility, hygiene, feeding, and medication adherence.

The economic value of informal caregiving was estimated using the replacement cost method, assigning the minimum wage in the informal sector as a proxy for the value of caregiver time. This approach was selected due to the high proportion of caregivers not formally employed or compensated for their support. No standardised wage was applied across participants; instead, reported costs represent opportunity costs rather than direct financial transactions.

All patients in the sample reported receiving informal care, eliminating the need for assumptions or imputations regarding caregiving status.

## Data analysis

Data entry was performed using Epi Info version 7.0 (Centers for Disease Control and Prevention, Atlanta, GA), and statistical analyses were conducted using SPSS version 20.0 (IBM Corporation, Armonk, NY, USA).

Descriptive statistics were used to summarize sociodemographic, clinical, and cost-related variables. Categorical variables were presented as frequencies and percentages, while continuous variables were summarized using means, medians, standard deviations (SD), interquartile ranges (IQR), and full ranges, as appropriate. Financial data, due to their typically skewed distribution, were primarily presented as medians and ranges [24].

All costs were calculated in Mozambican metical (MZN) for the year 2019 and converted to United States dollars (US$) using the average official exchange rate published by the Central Bank of Mozambique during the study period (US$1 = 61.02 MZN) [25].

Health-related quality of life (HRQoL) was assessed using EQ-5D-3L index scores. Although these scores are continuous, they were categorized into five levels to enhance interpretability and facilitate comparisons across patient subgroups. The categories were defined by the study team based on thresholds used in previous literature [26,27]: Very poor: < 0; Poor: 0.00–0.50; Fair: 0.51–0.85; Good: > 0.85 – < 1.00 e Perfect health: 1.00. This categorization was not based on a standardized classification system but was designed to support more informative interpretation of EQ-5D values in applied health research, particularly in settings where discontinuities in score distribution are common [27].

In addition to the index score, we analysed the EQ Visual Analogue Scale (EQ VAS), a 0–100 scale on which patients indicate their perceived health status on the day of assessment. The EQ VAS provides a complementary measure of overall health, reflecting the patient's self-perceived condition. Scores were summarized using means and standard deviations [28].

## Ethics statement

This study was approved by the Ethics Committee of the Faculty of Medicine and the Hospital Central de Maputo (Ref. No. 55/018). Written informed consent was obtained from all participants or their caregivers prior to data collection. The purpose and procedures of the study were explained in detail, and participants were informed of the voluntary nature of their involvement. All ethical procedures were conducted in accordance with the principles outlined in the Declaration of Helsinki.

## Results

### Population characteristics

In total 80 patients were admitted with a clinical diagnosis of stroke. Of these, 50 patients met the inclusion criteria and were recruited for the study, including 28 women (56%), all with a first-ever stroke event. The remaining 30 patients were excluded: 20 had a history of recurrent stroke, 4 were transferred to other facilities before recruitment, and 6 died prior to providing informed consent.

The mean age of participants was 60.9 years (interquartile range: 38.3–68.4 years), and 24 patients (48%) were younger than 60 years of age. Informal employment was reported by 32 participants (64%), and 21 (42%) reported earning less than the national minimum monthly wage of 4,390.00 MZN (approximately US$71.97) prior to the stroke event [29]. Additionally, 28 patients (56%) were identified as the primary income earners (householders) in their families. Further demographic and clinical characteristics are presented in Table 1.

### Clinical characteristics

Among the 50 patients included in the study, ischemic stroke was diagnosed in 28 patients (56%), while 22 patients (44%) presented with hemorrhagic stroke. The most common comorbidity was hypertension, which was present in all patients (100%). Other prevalent conditions included HIV infection (n = 10; 20%) and bronchopneumonia (n = 7; 14%).

**Table 1. Clinical characteristics by type of stroke.**

| Characteristics | All<br>N (%/IQR) | Ischemic (n = 28)<br>N(%) | Hemorrhagic (n = 22)<br>N(%) |
|---|---|---|---|
| Gender | | | |
| Male | 28 (56) | 16 (57.1) | 12 (54.6) |
| Female | 22 (44) | 12 (42.9) | 10 (45.5) |
| Mean age | 60.9 (38.3; 68.4) | 60,8 (38.2;69.1) | 60.9 (51,6;69,4) |
| Age group | | | |
| <60 years | 24 (48.0) | 15 (53.6) | 9 (40.9) |
| >60 years | 26 (52.0) | 13 (46.4) | 13 (59.1 |
| Family size | 5 (3.75; 7) | 5 (3.75; 7) | 5 (3.25; 7) |
| Disability | | | |
| Left | 23 (46.0) | 13 (46.4) | 10 (45.5) |
| Right | 22 (44.0) | 12 (42.9) | 10 (45.5) |
| Without | 5 (10.0) | 3 (10.7) | 2 (9.1) |
| Comorbidities | | | |
| HIV positive | 10 (20.0) | 8 (28.6) | 2 (9.1) |
| Bronchopneumonia | 7 (14.0) | 2 (7.1) | 5 (22.7) |
| Diabetes melitus | 1 (2.0) | 1 (3.6) | 0 |
| Arterial fibrillation | 1 (2.0) | 1 (3.6) | 0 |
| Aortic aneurism | 1 (2.0) | 0 | 1 (4.6) |
| Cardiomyopathy | 1 (2.0) | 1 (3.6) | 0 |
| Tuberculosis | 1 (2.0) | 0 | 1 (4.6) |
| Epilepsy | 1 (2.0) | 1 (3.6) | 0 |
| Risk factors | | | |
| Hypertension | 50 (100) | 28 (100) | 22 (100) |
| Family history of stroke | 7 (14.0) | 3 (10.7) | 4 (18.2) |
| Alcoholics habits | 13 (26.0) | 8 (28.6) | 5 (22.7) |
| Smoking | 8 (16.0) | 6 (21.4) | 2 (22.7) |
| Hospital stay<br>(median days) | 7 (14.0) | 6 (±3) | 8 (±3) |
| Discharge status (Alive) | 50 | 28 (56%) | 22 (44%) |
| Vital status after 28 days | | | |
| Alive | 40 (80%) | 23 | 17 |
| Death | 10 (20%) | 5 | 5 |

The median length of hospital stay was 8 days (IQR: 5–10) for patients with hemorrhagic stroke and 6 days (IQR: 4–8) for those with ischemic stroke. At discharge, 5 patients (10%) showed no motor deficits affecting the right or left side of the body.

Additionally, 10 patients (20%) died after hospital discharge but before the 28-day follow-up visit. Detailed clinical characteristics are presented in Table 1.

## Stroke admission cost

**Direct medical costs.** The direct hospital cost per patient was $US706.0 (43,069.08 MZN) from which $US278.62 (16,995.82MZN) for laboratory and diagnostic imaging, $US16.70 (1,018.72MZN) for medication and $US410.73 (25,054.54 MZN) for bed occupation Fig 2.

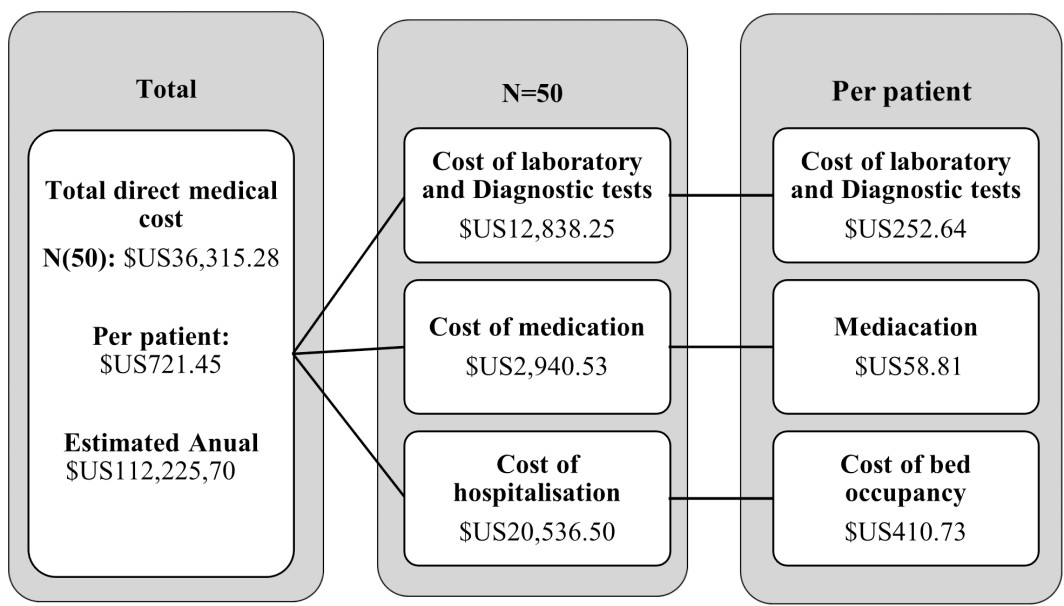

**Fig 2. Direct medical costs.**

This annual cost projection is based solely on the direct extrapolation of per-patient costs for first-ever strokes observed in this study. It does not account for recurrent stroke episodes, long-term care, or stroke cases managed outside the study setting. Therefore, while the figure offers a useful approximation of the annual burden, it should be interpreted with caution given the underlying assumptions.

**Direct nonmedical costs during hospitalization and 28 days post-discharge.** The estimated direct nonmedical cost per patient during hospitalization had a median value of US$12.59 (interquartile range [IQR]: 8.19–16.39) and a mean of US$13.62 (standard deviation [SD]: 8.02). At 28 days post-discharge, the median direct nonmedical cost per patient was US$32.04 (IQR: 19.01–49.83), and the mean was US$41.37 (SD: 36.11).

**Indirect cost.** Productivity lost per patient during hospitalization was US$ 87.21 and $16.93 respectively for those in formal and informal employment. This translates to $US523.28 (31,920.00MZN) for formal sector and $US541.64 (33,040.00 MZN) in the informal sector for the entire study cohort. The total indirect outpatient care costs were $US 1520.00 (91,200.00 MZN), with $US249.18 (15,199.98MZN) per patient for formal employment and $US1,229.17 (73,750.00 MZN ($US 1,229.17) with $US43.36 (2,644.96MZN) per patient for informal employment Table 2.

### Health-related quality of life

As shown in Fig 3, all five dimensions were seriously compromised in the 40 patients that survived 28 days after discharge Fig 3. Patients had more frequently (Level 2 and Level3) Anxiety/depression 37 (92.5%), incapacity for activities 30 (82.4%) and pain/discomfort 34(85.5%). The mean EQ-5D index, and 49.39 (SD, 20.95), respectively. Anxiety/depression 92.5% and Pain/discomfort 82.5% were the most frequently reported issues. The mean of VAS scores of stroke patients was 0.514 (SD, 0.298). These findings are detailed in Table 3, which summarizes the distribution of responses across each EQ-5D domain and the overall health utility measures.

### Discussion

This study provides new evidence on the economic and social burden of stroke in a low-income setting by evaluating both direct and indirect costs from the perspectives of the health system and the patient. The analysis incorporated data from

**Table 2. Indirect cost – loss of productivity (in-patient/out-patient).**

**I.Indirect cost**

**A. Indirect in-patient cost USD***

| Occupation | N | Monthly income | Daily income per patient | Income lost per patient | Total of income lost |
|---|---|---|---|---|---|
| Formal | 6 | 249.18 | 12.46 | 87.21 | 523.28 |
| Informal | 32 | 43.36 | 2.42 | 16.93 | 541.64 |
| Pensioners | 5 | 22.95 | 1.15 | 8.03 | 40.16 |
| Unemployed | 7 | 9.84 | 0.50 | 3.44 | 24.10 |
| Total | 50 | 340.16 | 16.52 | 115.60 | 1,129.18 |
| **B. Indirect out-patient care USD\*\*** | | | | | |
| Formal | 6 | 249.18 | 12.46 | 249.18 | 1,520.00 |
| Informal | 25 | 43.36 | 2.42 | 43.36 | 1,229.17 |
| Pensioners | 3 | 22.95 | 1.15 | 22.95 | 70.00 |
| Unemployed | 6 | 9.84 | 0.50 | 9.84 | 60.00 |
| Total | 40\*\*\* | 340.16 | 16.52 | 340.16 | 2,879.17 |
| Total | | | | | 3,961.16 |

*Note: we consider 5 working days for in-patient indirect cost.

**Note: loss of income due to absence from work was considered as 20 working days in the 28-day period.

***10 patients died before day 28 after discharge.

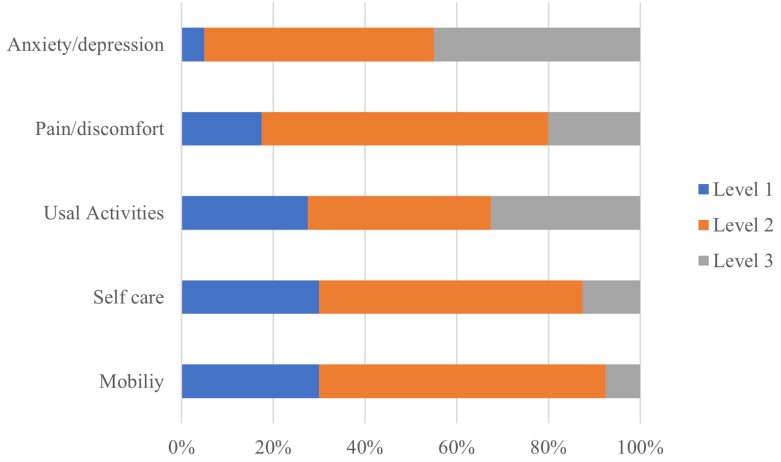

**Fig 3. Health-related quality of life on survivors from stroke in the first 28 days with: no problems (Level 1); some problems (Level 2); Extreme Problems (Level 3).**

hospitalization and community-based follow-up at 28 days post-discharge, capturing a broader view of the continuum of stroke care in Mozambique.

Approximately 70% of participants were from low socio-economic backgrounds, and 78% had only primary education or no formal schooling, reflecting the substantial vulnerability of the study population. These findings are consistent with other reports highlighting the association between low socio-economic status and limited access to health services, as well as poorer health outcomes in stroke patients [8].

**Table 3. Summary of EQ-5D measurement and index score.**

| | MOBILITY N (%) | SELF-CARE N (%) | USUAL ACTIVITIES N (%) | PAIN/DISCOMFORT N (%) | ANXIETY/DEPRESSION N(%) |
|---|---|---|---|---|---|
| Level 1 | 12 (30) | 12 (30) | 11 (27.5) | 6 (15) | 3 (7.5) |
| Level 2 | 25 (62.5) | 23 (57.5) | 16 (40) | 26 (65) | 20 (50) |
| Level 3 | 3 (7.5) | 5 (12.5) | 13 (32.5) | 8 (20) | 17 (42.5) |
| Total | 40 (100) | 40 (100) | 40 (100) | 40 (100) | 40 (100) |
| **EQ-5D Index score in four categories** | | | | | |
| Category | EQ-5D Index score | Frequency of patients N (%) | | | Mean (SD) |
| Very poor | <0 | 3 (7.5) | | | 0.514 (0.297) |
| Poor | 0 and 0.5 | 15 (37.5) | | | |
| Fair | 0.51 - 0.85 | 17 (42.5) | | | |
| Higher | >0.85 | 5 (12.5) | | | |
| Perfect health | 1 | 2 (5.0) | | | |

Follow-up assessments conducted at home revealed a lack of structured ambulatory care and minimal home-based support, underscoring persistent gaps in post-stroke rehabilitation services. These limitations likely contribute to prolonged disability, dependence on informal caregiving, and worsening economic hardship for affected households.

To the best of our knowledge, this is the first study in Mozambique to evaluate both the economic costs and health-related quality of life (HRQoL) of stroke survivors using standardized tools and including data from post-discharge follow-up. Previous studies in sub-Saharan Africa have mostly focused on clinical outcomes during hospitalization, with limited integration of economic impact or patient-centered measures such as HRQoL [10,21].

The direct cost per patient during hospitalization in an urban public health facility in this study was US$721.45, which aligns with cost estimates reported in other African settings. Across the continent, stroke-related hospitalization costs have been shown to range from approximately US$145 to US$4,860 per patient, depending on the level of care and resources available [7].

This wide variation reflects differences in healthcare facility type (public vs. private) and geographic context (urban vs. rural), with generally higher costs in urban and privately operated institutions due to more intensive diagnostic and treatment services [7]. However, studies assessing stroke-related costs in Africa remain scarce.. A study conducted in Togo estimated the direct cost per stroke patient at €936 for a 17-day hospitalization period. Using the 2015 average exchange rate (1 EUR = 1.11 USD), this corresponds to approximately US$1,039. This amount is nearly 170 times higher than the average annual per capita health expenditure in Togo, highlighting the disproportionate financial burden that stroke care imposes in low-resource settings [7]. The contrast between the short duration of hospitalization and the magnitude of the cost relative to national health spending underscores the urgent need for subsidized care models and risk-pooling mechanisms to protect vulnerable households from catastrophic health expenditures.

This finding also illustrates the immense economic burden that stroke care imposes in low-resource settings. Expanding and strengthening access to subsidized and well-structured post-discharge care may contribute to lowering mortality and morbidity rates across the region [7]. Regarding demographic characteristics, all patients in this study were Black, reflecting the demographic composition of the population served by the study hospital. The mean age at the time of stroke was 60.8 years, with nearly half of participants aged under 60 years. This finding aligns with global trends showing an increasing burden of stroke among adults younger than 65 years, particularly in low- and middle-income countries (LMICs), where the incidence of stroke in individuals aged 20–64 years increased by approximately 25% in recent decades [7,8].

In LMICs, the average age of stroke onset is notably younger estimated at around 64.4 years compared to high-income countries, where averages range from 71 years in Canada and the Netherlands to 74.9 years in Italy [8]. This difference partially reflects disparities in life expectancy, access to preventive care, and chronic disease management between LMICs and high-income countries.

The need for long-term care following stroke poses a considerable human and economic burden, especially in health systems with limited rehabilitation services. Furthermore, individuals living in socioeconomically deprived conditions are disproportionately affected not only by a higher prevalence of risk factors such as hypertension and HIV, but also by more severe stroke presentations, earlier onset, and higher mortality rates compared to their more affluent counterparts [8].

The younger average age of stroke onset observed in this study (mean 60.8 years) is consistent with trends in low- and middle-income countries (LMICs), where stroke increasingly affects adults under the age of 65 [7,8]. Survival rates tend to be higher among younger stroke patients [30], which often results in long-term caregiving responsibilities for cohabiting relatives, who may support the survivor for many years. This dynamic contributes to sustained social and financial strain on households, particularly when the stroke affects the primary income earner.

Several factors may help explain the earlier onset of stroke in LMICs. First, there is a high prevalence of uncontrolled hypertension, frequently undiagnosed or inadequately treated due to limited access to screening and continuity of care [31,32]. Second, infectious diseases, particularly HIV/AIDS, may independently increase stroke risk through mechanisms such as vasculitis, opportunistic infections, and cardioembolic complications [33,34]. Third, socioeconomic hardship, chronic stress, and poor nutrition further amplify cardiovascular risk in younger adults [35,36]. Additionally, low health literacy is associated with reduced uptake of preventive strategies and delayed management of risk factors [37].

These combined influences may account for the earlier stroke onset observed in Mozambique relative to high-income countries, where more robust health systems allow for better risk factor control and earlier medical intervention. The interplay between noncommunicable diseases and poverty is particularly evident in this context, reinforcing a cycle in which chronic illness impairs economic productivity, while poverty exacerbates disease severity and limits access to care [8]. Hypertension was the most prevalent risk factor in our cohort, affecting 100% of participants, consistent with its role as the leading modifiable risk factor for stroke worldwide [5]. In Mozambique, the high prevalence of uncontrolled hypertension is exacerbated by limited access to routine blood pressure screening, inadequate long-term follow-up, and poor availability of antihypertensive medications in public health facilities [31,32]. These barriers contribute to higher stroke incidence and severity across the population.

In addition to hypertension, HIV infection, observed in 20% of participants, represents a major comorbidity that significantly affects stroke outcomes. Mechanisms such as HIV-associated vasculopathy, chronic systemic inflammation, and opportunistic infections have been linked to both ischemic and hemorrhagic stroke subtypes [33,34]. This dual burden of communicable and non-communicable diseases presents a significant challenge for healthcare systems in Mozambique and other sub-Saharan African countries, underscoring the need for integrated care models that address both disease categories simultaneously.

Globally, it is estimated that 3–4% of total healthcare expenditures are devoted to stroke care, based primarily on data from high-income countries with established health financing systems [3,38]. In our study setting, the stroke-related expenditures at Hospital Geral de Mavalane in 2019 also accounted for approximately 3% of the hospital's total annual budget, which appears to mirror global estimates. However, in the context of limited national resources, the impact of stroke is disproportionately greater, given the high societal costs, younger age of onset, and limited rehabilitation services.

In 2019, the Mavalane Geral Hospital total budget was approximately US$4,436,142, with stroke care accounting for 3% of this amount consistent with global averages. However, despite similar budgetary proportions, outcomes in Mozambique are considerably worse, reflecting the challenges of delivering effective care in resource-constrained settings. This discrepancy underscores the urgent need to improve the efficiency and impact of current spending through better prevention, early detection, and integrated care strategies tailored to the local context.

Several studies conducted across Africa have consistently reported that stroke survivors experience poorer health-related quality of life (HRQoL) across multiple domains when compared to stroke-free controls [10,21]. The most frequently identified predictors of reduced HRQoL include post-discharge disability, depression, and stroke severity. In our study, 80% of stroke survivors reported symptoms of depression and anxiety at 28 days post-discharge. This finding is aligned with previous research in low-resource settings, where psychological distress is compounded by functional limitations, loss of income, and limited access to rehabilitation and mental health services [10,39].

In Mozambique, as in many LMICs, mental health services are scarce and often not integrated into post-stroke care pathways. Factors such as socioeconomic stress, caregiver burden, and low health literacy further exacerbate the risk of untreated mental health conditions. These observations underscore the need for comprehensive post-discharge strategies that incorporate both physical rehabilitation and psychosocial support [37,40].

The economic burden of stroke, coupled with diminished HRQoL and limited access to services, poses a significant public health challenge in Mozambique. The impact is especially severe given that stroke increasingly affects individuals in their productive working years [8]. When a young or middle-aged adult suffers a stroke, households face the loss of income, long-term caregiving responsibilities, and escalating financial strain. In our cohort, the mortality rate of 20% within 28 days post-discharge further highlights the severity of stroke outcomes in this setting and the fragility of family-based support systems in the face of early loss of a primary income earner.

Socioeconomic deprivation has a documented role not only in increasing the prevalence of stroke risk factors but also in intensifying stroke severity, mortality, and early onset [8]. Individuals from disadvantaged backgrounds often have reduced access to healthcare, poorer control of chronic conditions, and limited capacity for lifestyle modification. This vulnerability is particularly evident among younger stroke survivors, who, despite higher survival rates [30], become dependent on family caregivers for prolonged periods, placing additional strain on household resources.

Analysis of income and expenditure data in our study revealed that stroke care costs often exceeded monthly household income, particularly for informal workers and pensioners. For example, informal workers spent approximately 139% of their monthly earnings on stroke-related expenses, indicating a catastrophic financial burden. These findings call attention to the need for financial protection mechanisms and targeted social support programs to mitigate the economic impact of stroke on vulnerable households.

Our follow-up also revealed that the lack of structured ambulatory and home-based care further intensifies the long-term burden of stroke in Mozambique. Most patients were discharged without access to rehabilitation services, physiotherapy, or community-based health support. This discontinuity in care contributes to prolonged disability, increased complication rates, and greater reliance on unpaid, informal caregiving by family members.

In contrast, high-income countries have implemented post-discharge rehabilitation and home-based care as core components of stroke recovery, contributing to better functional outcomes and reduced hospital readmissions [41,42]. The absence of these services in LMICs perpetuates a cycle of poor recovery, economic hardship, and systemic healthcare strain [2,43,44]. Therefore, strengthening community-level rehabilitation infrastructure and integrating mental health support into stroke care pathways should be a policy priority in Mozambique and similar low-resource settings.

## Limitations

Despite the strength of our study, there are some limitations that need to be acknowledged. First, its relatively small sample size (n = 50) and single-center design may limit generalizability. However, efforts were made to recruit consecutively and systematically to reduce selection bias.

Second, the 28-day follow-up captures only short-term outcomes and costs. While this provides insight into early post-discharge burden, it does not reflect longer-term rehabilitation or recurrence, which should be addressed in future studies.

Third, although individual income data were collected, total household income was not, potentially underestimating financial impact. Still, income was cross-checked with employment status and wage references to improve reliability.

Fourth, cost data relied on patient and caregiver self-reports, which may introduce recall bias. To mitigate this, home visits were conducted to validate responses and clarify inconsistencies.

Fifth, the EQ-5D-3L instrument was used with a Zimbabwean value set due to the lack of a Mozambican-specific tariff. While this may affect precision, it offers the best available regional proxy for health utility estimation.

Finally, patients with recurrent stroke were excluded to avoid confounding related to pre-existing disability. This may underestimate total system burden, but it improved comparability and consistency in first-event cost estimation.

Despite these limitations, the study applied rigorous methods and multiple validation steps, offering robust evidence to inform stroke care and policy in low-resource settings.

## Conclusion

Stroke is one of the major public health problems due to its morbidity, mortality, disability, and its economic impact. This present study shows that the cost of Stroke hospitalization is significant for the health system and for the patient, the loss of productivity due to stroke is high taking in account the low economic status and the age of the affected people in our context. There is limited ambulatory and home-based care support for the patient who survived stroke.

The health-related quality of life of the surviving patients are very low in first 28 days after hospital discharge with significant prevalence of depression. There is the need for a primary prevention of the risk factors of this pathology and the effective implementation of universal health coverage. For future work we intend to continually understand the phenomenon.

## Acknowledgments

The authors would like to thank MIHER (Mozambican Institute for Health Education and Research), the University of Eduardo Mondlane, Faculty of Medicine, the clinical personnel of the Hospital Geral de Mavalane and Instituto Nacional de Saúde.

## Author contributions

**Conceptualization:** Igor Samuel Dobe.

**Data curation:** Igor Samuel Dobe.

**Formal analysis:** Igor Samuel Dobe, Ana Mocumbi.

**Investigation:** Igor Samuel Dobe.

**Methodology:** Igor Samuel Dobe, Neide Canana, Simon Stewart.

**Resources:** Igor Samuel Dobe.

**Supervision:** Neide Canana, Clifford Afoakwah, Simon Stewart, Ana Mocumbi.

**Validation:** Clifford Afoakwah, Simon Stewart, Ana Mocumbi.

**Visualization:** Neide Canana, Simon Stewart, Ana Mocumbi.

**Writing – original draft:** Igor Samuel Dobe.

**Writing – review & editing:** Neide Canana, Ana Mocumbi.

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
