## [Decision Letter · Decision Letter 0]

PONE-D-23-21542The cost and quality of life of stroke-related management and care in Mozambique: A prospective observational cohort study of acutely hospitalized casesPLOS ONE

Dear Dr. Dobe,

Thank you for submitting your manuscript to PLOS ONE. After careful consideration, we feel that it has merit but does not fully meet PLOS ONE’s publication criteria as it currently stands. Therefore, we invite you to submit a revised version of the manuscript that addresses the points raised during the review process.

Please carefully review and respond to the reviewer's very constructive comments, including the additional methodological details required to fully judge veracity of the analysis and results presented in this manuscript. 

We look forward to receiving your revised manuscript.

Kind regards,

Ryan G Wagner, MSc(Med), MBBCh, PhD

Academic Editor

PLOS ONE

http://etd.aau.edu.et/bitstream/handle/123456789/30077/Kemal%20Ali.pdf?isAllowed=y&sequence=1

https://bmcmedicine.biomedcentral.com/articles/10.1186/s12916-019-1397-3

https://www.nature.com/articles/s41582-021-00542-4?code=a6c15553-581a-4f47-8809-7829152f121a&error=cookies_not_supported

In your revision ensure you cite all your sources (including your own works), and quote or rephrase any duplicated text outside the methods section. Further consideration is dependent on these concerns being addressed.

4. We suggest you thoroughly copyedit your manuscript for language usage, spelling, and grammar. If you do not know anyone who can help you do this, you may wish to consider employing a professional scientific editing service. 

A clean copy of the edited manuscript (uploaded as the new *manuscript* file)”.

 [No - The funders had no role in study design, data collection and analysis, decision to publish, or preparation of the manuscript.]. 

6. In the online submission form, you indicated that your data will be submitted to a repository upon acceptance.  We strongly recommend all authors deposit their data before acceptance, as the process can be lengthy and hold up publication timelines. Please note that, though access restrictions are acceptable now, your entire minimal  dataset will need to be made freely accessible if your manuscript is accepted for publication. This policy applies to all data except where public deposition would breach compliance with the protocol approved by your research ethics board. If you are unable to adhere to our open data policy, please kindly revise your statement to explain your reasoning and we will seek the editor's input on an exemption. 

Reviewers' comments:

Reviewer's Responses to Questions

**Comments to the Author**

1. Is the manuscript technically sound, and do the data support the conclusions?

Reviewer #1: Partly

Reviewer #2: Partly

2. Has the statistical analysis been performed appropriately and rigorously? 

Reviewer #1: No

Reviewer #2: No

3. Have the authors made all data underlying the findings in their manuscript fully available?

Reviewer #1: No

Reviewer #2: No

4. Is the manuscript presented in an intelligible fashion and written in standard English?

Reviewer #1: Yes

Reviewer #2: No

5. Review Comments to the Author

Reviewer #1: With a sample size of 50 patients from a single urban hospital, the study's findings might have limited generalizability to the wider Mozambican population or other low-income countries. The manuscript does not sufficiently address this limitation or discuss how the findings might or might not reflect the situation in different settings within Mozambique or similar countries.

The manuscript does not sufficiently detail the selection process for the study's participants or how potential biases were mitigated. For a study of this nature, these aspects are key to validate the findings.

A comparison with findings from other regions, especially those with similar socioeconomic settings, would have provided a richer understanding of the study's implications.

Reviewer #2: Abstract

1) Results. What do the authors mean by the “informal sector”? This has not been presented anywhere else in the abstract.

2) Results. It would be good to report the EQ-5D results in the abstract rather than just a narrative summary.

3) Results. Possible typo? The IQR of $13,558 cannot be 4,3 to 7,42.

4) Conclusion. The authors should rewrite this section. What do they mean by the cost implications … is particularly high? What is the government/health care sector to do? In addition, strokes might occur at younger ages in Mozambique than say in Western Europe, but their life expectancy is also generally lower.

Methods

1) Line 109. Details have to be provided of what the human capital approach entails.

2) Line 120. The authors collected income data from patients (line 99). Why was this information not used rather than the minimum wage?

3) Line 132. Why was the 3L version used over the 5L?

4) Line 132. Which valuation set, and from which country, was used to value responses to the EQ-5D?

5) Line 139. Why were not means not presented too? Means might be useful to policy makers to establish overall costs of a disease (i.e. mean cost of stroke X number of strokes). Median costs do not allow for these calculations.

6) Line 144. Why were summary measures of EQ-5D not presented? Related to this, how did the authors come up with the EQ-5D groupings?

Results

1) Line 161. Why were the other 30 patients not recruited? Were not incident strokes, refused, etc…?

2) Line 163. I did see no mention of formal/informal employment in the methods. What do they mean by informal employment?

3) Line 165. Details of the minimum wage should be given in the methods. What does the minimum wage use represent: per day, per week, per month?

4) Table 1. Some elements of the table are not aligned (i.e. gender).

5) Table 1, line 162. Why does the table report mean age whilst the text reports median age?

6) Table 1. Some abbreviations in the table need spelling out (e.g. HTN for hypertension).

7) Line 174. The authors mention that 5 patients had minor limitation, whereas the table only reports no, left or right limitation.

8) Paragraph starting line 185. 4 costs are presented, one is the total, and two others representing transport and medication. What did the other cost relate to?

9) Paragraph starting line 189. A) Were these total costs or costs per patient? B) Either costs are reported in US $ first or in MZN. C) What were indirect outpatient care costs?

10) Table 2. Much of the details in Table 2 should be reported in the methodology.

11) Figure 2. I’m not sure how results of the EQ-5D are reported in Table 2. Are these the people who said I have some or extreme problems in each of the 5 domains?

12) In the methods the authors reported that EQ-5D results were presented in 4 groups. No such data are presented in the results.

Discussion

1) Line 229. The authors appear to imply that having a stroke young, as is the case in Mozambique with average age at first stroke being 60 years, will result in an enormous burden for the years to come. However, I note from the World Bank that the average life expectancy in Mozambique is 61.17 years, more or less the age at which patients are having their strokes. Therefore, I do not think this line of argument hold. By contrary, Italy where the authors quote estimates of 74.9 years as the age of first stroke, life expectancy is 82.34 years.

2) Limitations. I found this section very short. Chief amongst the limitations of this study is the small patient sample size. Also, how generalizable are their results to the rest of the country? For example, I note that the study was based on a tertiary hospital in the capital.

6. PLOS authors have the option to publish the peer review history of their article (what does this mean? ). If published, this will include your full peer review and any attached files.

**Do you want your identity to be public for this peer review?** For information about this choice, including consent withdrawal, please see our Privacy Policy .

Reviewer #1: No

Reviewer #2: **Yes: ** Ramon Luengo-Fernandez

---

## [Author Response · Author response to Decision Letter 1]

26 Aug 2024

Comments from Reviewer 1

1) With a sample size of 50 patients from a single urban hospital, the study's findings might have limited generalizability to the wider Mozambican population or other low-income countries. The manuscript does not sufficiently address this limitation or discuss how the findings might or might not reflect the situation in different settings within Mozambique or similar countries.

Response: Thanks a lot for this comment, there is very few studies in sub-Saharan Africa assessing the cost of stroke management and/or the recovery trajectory of health-related quality of life after a stroke event, However, we have added more information and references in the discussion regarding how the findings reflect the findings in similar countries.

2) The manuscript does not sufficiently detail the selection process for the study's participants or how potential biases were mitigated. For a study of this nature, these aspects are key to validate the findings.

Response: We have included more details regarding to the selection process and the bias mitigation process: “Eligible participants were identified through daily hospital registries to avoid selection bias”. Please see lines 93-102.

3) A comparison with findings from other regions, especially those with similar socioeconomic settings, would have provided a richer understanding of the study's implications.

Response: Thanks a lot for this comment, we have now included more information and references in the discussion regarding how the findings reflect the findings in similar countries. (comment similar with the first comment). Please see line 264-279.

Comments from Reviewer 2

Abstract

1) Results. What do the authors mean by the “informal sector”? This has not been presented anywhere else in the abstract.

Response: Thanks for this comment we have included more information regarding to informal sector and we have reworded it as informal employment to make clearer throughout the manuscript. “Loss of productivity due to stroke which varied by employment status (informal, formal, pensioner and unemployed)”. Please see line 40. Formal and informal employments have also been defined on lines 107-111.

2) Results. It would be good to report the EQ-5D results in the abstract rather than just a narrative summary.

Response: We have added this information in the abstract: “The mean EQ-5D index and VAS scores of stroke patients were 0.514 (SD, 0.298), and 49.39 (SD, 20.95), respectively. Anxiety/depression 82.5% and Pain/discomfort 72.5% were the most frequently reported issues.” See line 54.

3) Results. Possible typo? The IQR of $13,558 cannot be 4,3 to 7,42.

Response: Thanks for raising this point, it was typo and we have corrected it accordingly on line 49

4) Conclusion. The authors should rewrite this section. What do they mean by the cost implications … is particularly high? What is the government/health care sector to do? In addition, strokes might occur at younger ages in Mozambique than say in Western Europe, but their life expectancy is also generally lower.

Response: We have rewritten the conclusions in the abstract as recommended. The revision now reads: “The economic cost of stroke management in low-income sub-Saharan African countries such as Mozambique is substantially high, with considerable out-of-pocket spending, poor survival rate and a compromised health-related quality-of-life. Health system reforms designed to mitigate the individual to societal burden imposed by stroke are required.” Please see line 58

Methods

1) Line 109. Details have to be provided of what the human capital approach entails. Response: We provided more details regarding the human capital approach “that is, the value of lost productive time due to acute illness and short-and-long-term disabilities” Please see line 122.

Reference: J. Pike and S. D. Grosse, “Friction Cost Estimates of Productivity Costs in Cost-of-Illness Studies in Comparison with Human Capital Estimates: A Review,” Applied Health Economics and Health Policy, vol. 16, no. 6. Springer International Publishing, pp. 765–778, Dec. 01, 2018. doi: 10.1007/s40258-018-0416-4. Please sea line 113.

2) Line 120. The authors collected income data from patients (line 99). Why was this information not used rather than the minimum wage?

Response: Thanks a lot for this recommendation, we agree, and have used the income data from the patients instead of minimum wage to compute lost productivity. Please see line 143

3) Line 132. Why was the 3L version used over the 5L?

Response: Thanks for this question. The EQ-5D-3L was selected instead of the EQ-5D-5L due to unavailability of the EQ-5D-5L in Portuguese version at the time designed the study. Please see line 159

4) Line 132. Which valuation set, and from which country, was used to value responses to the EQ-5D?

Response: Thanks a lot, we specified the valuation set we used: “There is no value of set for EQ-5D for both 5L and 3L for Mozambique, because of that we used the coefficients from published 3L version TTO (time trade-off) valuation study from Zimbabwe.” Please see line 160

Reference: Jelsma J, Hansen K, De Weerdt W, De Cock P, Kind P. How do Zimbabweans value health states? Popul Health Metr. 2003 Dec 16;1(1):11. doi: 10.1186/1478-7954-1-11. PMID: 14678566; PMCID: PMC317383.

5) Line 139. Why were not means presented too? Means might be useful to policy makers to establish overall costs of a disease (i.e. mean cost of stroke X number of strokes). Median costs do not allow for these calculations.

Response: Thanks for this comment, we agree, and have included means on the analysis and results. Please see line 167

6) Line 144. Why were summary measures of EQ-5D not presented? Related to this, how did the authors come up with the EQ-5D groupings?

Responses: We have included summary measures of EQ-5D as recommended. Please see line 166 and Table 3.

Results

1) Line 161. Why were the other 30 patients not recruited? Were not incident strokes, refused, etc…?

Response: We have now clarified this in the results section. Specifically, have added on line 170 that “In total 80 stroke patients were admitted during the study period, of those, 50 were recruited (28 women, 56%) with diagnoses of first in life stroke incident, other 30 participants didn’t meet the inclusion criteria’s (20 recurrent stroke, 4 were transferred and 6 died before the informed consent).” See Line 194

2) Line 163. I did see no mention of formal/informal employment in the methods. What do they mean by informal employment?

Response: We have now included the definition of formal/informal employment in the methods section as “informal employment’ refers to employment, which is not regulated, or which does not enjoy core social protections, formal employment is regulated with social protections” on line 115

3) Line 165. Details of the minimum wage should be given in the methods. What does the minimum wage use represent: per day, per week, per month?

Response: The minimum wage represents the one-month salary we included more details in the manuscript. Please see line 199

4) Table 1. Some elements of the table are not aligned (i.e. gender).

Response: Table revised, and elements aligned. Please Table 1.

5) Table 1, line 162. Why does the table report mean age whilst the text reports median age?

Response: We have standardized age using mean age in both table and text. Please see line 197

6) Table 1. Some abbreviations in the table need spelling out (e.g. HTN for hypertension).

Response: Abbreviated word has been spelt out as recommended in the table. Please see Table 1

7) Line 174. The authors mention that 5 patients had minor limitation, whereas the table only reports no, left, or right limitation.

Responses: Thank you for these comments, we have clarified this in the text. See Line 209

8) Paragraph starting line 185. 4 costs are presented, one is the total, and two others representing transport and medication. What did the other cost relate to?

Response: We have included information regarding to the other cost on the paragraph “The total cost for the patient was $US61.55 (2,461.80MZN) corresponding to: medication $US17.85 (1,089.00MZN), transport to and from hospital $US14.13(862.05MZN) and other costs $US8.37 (510.75.MZN).” please see line 220.

9) Paragraph starting line 189. A) Were these total costs or costs per patient?

Response: This is costs per patient and have been clarified on line 229.

B) Either costs are reported in US $ first or in MZN.

Response: Thanks for rising this point, all costs have been reported in US$ with MZN reported in brackets. See line 225

C) What were indirect outpatient care costs?

Response: We have included more information related to outpatient costs: “The total Indirect outpatient care costs were $US1,520.00 (91,200.00 MZN), with $US249.18(15,199.98MZN) per patient for formal employment and $US1,229.17 (73,750.00MZN) with $US43,36 (2,644.96MZN) per patient for informal employment”. Please see line 232

10) Table 2. Much of the details in Table 2 should be reported in the methodology.

Response: Thanks for this comment, we have revised the method to include details of Table as recommended. Please see line 112

11) Figure 2. I’m not sure how results of the EQ-5D are reported in Table 2. Are these the people who said I have some or extreme problems in each of the 5 domains?

Response: Thank you for this comment, we have improved the graphic to include all 3 levels of the 5 dimensions. Please see figure 328

12) In the methods the authors reported that EQ-5D results were presented in 4 groups. No such data are presented in the results.

Response: We have now reported the EQ-5D results in the table 3

Discussion

1) Line 229. The authors appear to imply that having a stroke young, as is the case in Mozambique with average age at first stroke being 60 years, will result in an enormous burden for the years to come. However, I note from the World Bank that the average life expectancy in Mozambique is 61.17 years, the age at which patients are having their strokes. Therefore, I do not think this line of argument hold. By contrary, Italy where the authors quote estimates of 74.9 years as the age of first stroke, life expectancy is 82.34 years.

Response: Thanks for this comment, we have revised our arguments regarding to age of stroke. Please se line 286

2) Limitations. I found this section very short. Chief amongst the limitations of this study is the small patient sample size. Also, how generalizable are their results to the rest of the country? For example, I note that the study was based on a tertiary hospital in the capital.

Response: We have included more information in this section: “Despite the strength of our study, there are some limitations that need to be acknowledged. Firstly, unlike many advanced countries with structured administrative hospital and cost data, no detailed cost structure is attributed to care provided in Mozambique health facilities for the financial management of medical and paramedical activities. Also, the unit costs charged by the national health service are highly subsidized. Hence, the estimates in this study are patient-reported and could be subject to recall bias. Secondly, this study was implemented in only one tertiary health facility with limited number of patients being admitted during the study period, hence, our findings should be interpreted with caution.” Please see line 316.

---

## [Decision Letter · Decision Letter 1]

PONE-D-23-21542R1The cost and health-related quality of life of stroke management and care of acutely hospitalized cases in MozambiquePLOS ONE

Dear Dr. Dobe,

Thank you for submitting your manuscript to PLOS ONE. After careful consideration, we feel that it has merit but does not fully meet PLOS ONE’s publication criteria as it currently stands. Therefore, we invite you to submit a revised version of the manuscript that addresses the points raised during the review process.

I would ask the authors to carefully review and respond to the comments raised by the reviewer. Kindly provide a point-by-point response.  Please submit your revised manuscript by May 15 2025 11:59PM. If you will need more time than this to complete your revisions, please reply to this message or contact the journal office at plosone@plos.org . Please include the following items when submitting your revised manuscript:

We look forward to receiving your revised manuscript.

Kind regards,

Ryan G Wagner, MSc(Med), MBBCh, PhD

Academic Editor

PLOS ONE

Reviewers' comments:

Reviewer's Responses to Questions

**Comments to the Author**

1. If the authors have adequately addressed your comments raised in a previous round of review and you feel that this manuscript is now acceptable for publication, you may indicate that here to bypass the “Comments to the Author” section, enter your conflict of interest statement in the “Confidential to Editor” section, and submit your "Accept" recommendation.

Reviewer #3: (No Response)

2. Is the manuscript technically sound, and do the data support the conclusions?

Reviewer #3: Partly

3. Has the statistical analysis been performed appropriately and rigorously? 

Reviewer #3: I Don't Know

4. Have the authors made all data underlying the findings in their manuscript fully available?

Reviewer #3: Yes

5. Is the manuscript presented in an intelligible fashion and written in standard English?

Reviewer #3: No

6. Review Comments to the Author

Reviewer #3: Firstly, I would like to thank the authors for their valuable contribution to the body of knowledge on stroke research in Africa.

Abstract (Lines from Second draft of manuscript unless stated otherwise).

1. Line 37 - The authors mention that health-related quality of life was assessed at 28 days post-stroke using the EQ-5D-3L questionnaire, but in line 138, it’s stated as 28 days post-discharge. Could you please clarify which timeline is correct?

Introduction

2. Lines 65-66 - The authors state that stroke is now among the top ten causes of death in Mozambique, with a 25% increase between 2009 and 2019. Could you clarify from what position to what position stroke moved in the ranking of causes of death during this period?

3. Lines 67–81 - the rationale for conducting a cost analysis is articulated, with the focus being on economic burden and resource allocation in Mozambique. However, there is no similar argument presented for assessing quality of life at 28 days post-discharge. Is the quality-of-life assessment intended to contribute to the societal perspective analysis, and if so, could you provide more context on why this was an important outcome to measure in this study?

Materials and Method

4. Line 86 - Given the small sample size (n=50) and the single-hospital setting, I recommend revising the term ‘representative cohort’ to more accurately reflect the study’s context and limitations. A possible alternative could be: ‘a cohort of first-ever stroke patients admitted to a tertiary-referral hospital…,’ which would appropriately manage the readers’ expectations regarding the study’s generalisability.

5. There seems to be an inconsistency between lines 33 and 100-101 regarding the data collection method. In line 33, it is stated that indirect costs were derived from ‘structured interviews,’ but in the methods section (lines 100-101), it is mentioned that data were actually collected using a ‘semi-structured questionnaire.’ Please consider revising for consistency, these are not the same thing. Also is it possible to see what this questionnaire looks like?

6. Lines 106 -108 - The authors mention that formal employment is regulated with ‘social protections.’ Could you clarify what types of social protections this includes, especially considering that these may differ in Mozambique compared to other countries?

7. Line 117 - The authors mention caregiver costs as part of the direct and indirect costs 28 days post-discharge. Could you clarify if the cost of caregivers was standardised, and how you quantified this cost if a family member served as the caregiver? Additionally, was there an assumption made about caregiver costs for patients who couldn’t afford one, or did all participants have formally employed caregivers, or were informal caregiving arrangements (family, friends, etc.) also included?

8. Line 139 – typo; a multidimensional concept ‘which’ incorporates…

9. Lines 158 –165 – The authors describe the categorisation of EQ-5D index scores into five levels of HRQoL to facilitate interpretation and statistical modelling. Could the authors please clarify if these categories (< 0 ‘very poor,’ 0-0.5 ‘poor,’ 0.51-0.85‘fair,’ > 0.85 ‘good,’ and 1 ‘perfect state’) were custom defined for this study or if they were based on any previously validated or published classification system in the literature? How did they come up with these?

10. Figure 1 – typo; were there 9 deaths or 10 deaths?

Results

11. Table 1 – Misaligned gender and disability values

12. Line 198 – I was wondering if the costs of care for the comorbidities were separated from the data collected at 28 days post-discharge, or if they were combined into a single cost estimate. Could this potentially introduce confounding effects if all costs were lumped together?

13. Figure 2 - the ‘Estimated Anual’ cost of $US112,225.70 appears to be derived from the per-patient cost for first-time strokes. It’s interesting to see this estimate included without a discussion of its assumptions, such as excluding recurrence and the limitations of the number of stroke cases. Wouldn’t this estimate need to account for the costs of recurrent strokes as well to provide a more accurate reflection of the annual burden? Additionally, does this mean that the estimate would hold if we were to assume that a similar proportion of first-time stroke cases relative to total stroke cases occurs each year?

14. Lines 213-219 and Table 2 - In Line 131 the authors mention that indirect costs included the calculation of days of lost work to assess productivity loss. Since the day-to-day income of informally employed individuals often fluctuates, could you clarify if the income used in the cost assessment was an average calculated across all patients, or if another approach was used to account for income variability?

15. In Table 2 - The authors provide data on monthly income and the total cost of care, but it might be helpful to include as well what percentage of each income group’s monthly income is spent on stroke care costs. This could give a clearer picture of the financial burden on different social classes. Would this analysis be possible to add?

Discussion

16. Line 245 – typo; ‘regarding to the author’s knowledge’

17. Line 251 – the cost in Togo is reported in Euros, but the rest of the article uses US dollars. For consistency, could this be converted to US dollars?

18. Lines 254–262, the authors note that stroke affects younger people in Mozambique and other developing countries (<65 years) but don’t discuss why this might be the case. It would be helpful to explore possible contributing factors, such as a higher burden of infectious diseases (e.g., HIV) etc., rather than just comparing to age statistics from high-income countries without critique.

19. Line 254- Still on demographic characteristics; the authors don’t talk at all about hypertension, which was the most common risk factor, or HIV, the most common comorbidity among their participants. It seems like a big gap not to discuss what these conditions mean for stroke outcomes and costs in Mozambique.

20. In table 1 the authors mention that 10 (20%) of participants died within 28 days of discharge. What might this mean for the families of these patients, especially if the deceased was the primary income earners? It would be helpful to know the author’s thoughts on the potential social and economic impact of these premature deaths on affected households.

21. The discussion doesn’t flow well as some points. For example, it jumps from talking about socioeconomic deprivation (line 261) to survival rates (line 263) in younger people without any clear link between these points. The sentences feel disconnected, making it hard to follow the argument.

22. Lines 264-268 - When the authors mention the financial impact of stroke on families, they don’t provide any specifics, like what percentage of the family’s income is spent on stroke care. Without that kind of detail, it’s hard to judge how serious the financial burden really is.

23. I was wondering if the authors collected any data on the total family income of the participants in this study apart from the income per-patient. If so, it would be helpful to see how the costs of care compare to total family income.

24. Lines 269–276, the authors state that stroke costs were 3% of the budget, matching global estimates. However, the conclusion that this money could be better spent on other diseases or prevention feels out of place and doesn’t follow logically from the rest of the paragraph. A more relevant point might have been to explore why Mozambique is spending a similar percentage to other countries but achieving worse outcomes.

25. Line 271 – What does HGM stand for? Please rather write it in full.

26. Lines 281–282 – The authors mention that 80% of survivors had depression and anxiety, in line with other studies in Africa. The authors should consider discussing what they think might be causing this, and what the literature says about the possible contributing factors such as access to mental health services or socioeconomic stress?

27. Lines 283–285 - The concluding paragraph feels unclear and difficult to follow. The connection between the high cost of managing stroke, poor quality of life, and the prognosis for younger, working-age populations could be articulated more clearly. It might help to rephrase this section

Limitations

28. Line 182 – Could the exclusion of the 20 patients with recurrent stroke introduce potential selection bias in the data? What are the implications of excluding these patients for your study’s findings? Additionally, what percentage of strokes typically go on to be recurrent in Mozambique?

Conclusion

29. Lines 304 – 305 – The authors mention that there is limited ambulatory and home-based care support for stroke survivors. However, since this point wasn’t expanded upon in the discussion, it feels somewhat out of place here. Would it be possible to either expand on this point in the discussion or clarify its relevance in the conclusion?

My biggest concern with this article is in the introduction and discussion sections, where there are issues of inadequate referencing, identical wording to original sources, and claims that are not supported by the cited references.

Inadequate referencing

Line 60 - “Ranking as the second leading cause of death worldwide, with 5.8 million fatal cases per year (Kocarnik et al., 2022).”

• The Kocarnik et al. (2022) article does not mention stroke as the second leading cause of death or provide the figure of 5.8 million fatal cases per year at all. Instead, it focuses on cancer statistics. The source cited does not support the claim made in the manuscript. Additionally, the most recent Global Burden of Disease study (2024) has been published with relevant stroke figures which the authors could have used.

Line 65-66 - “It is now among the top ten causes of death, with a 25% increase between 2009 and 2019 (Bukhman, Mocumbi, and Horton, 2015)”

• The Bukhman, Mocumbi, and Horton (2015) article does not mention stroke as one of the top ten causes of death or provide a 25% increase figure at all. It focuses more broadly on the burden of non-communicable diseases.

Lines 269-270 - “Global estimates show that 3 to 4% of total health care system resources are devoted to stroke (21)”.

• The cited source (Rits IA, Declaration of Helsinki, 1964) is not appropriate for this claim, and does not contain such figures

Discussion

Identical wording to original sources

Lines (215-217 in the First draft of manuscript) - “US$ 145 to US$ 4,860, depending on the care setting. Cost of stroke care is higher in urban areas than in rural areas and higher in private health facilities than in government health facilities, (Akinyemi et al., 2021)”

• Actual Akinyemi et al. (2021) article words under subsection Quality of life and cost of care – “The estimated cost of care per patient with stroke ranges from US$ 145 to US$ 4,860, depending on the care setting. Cost of stroke care is higher in urban areas than in rural areas and higher in private health facilities than in government health facilities.” The phrasing is nearly identical without paraphrasing or quotation marks in the manuscript.

Lines 250 – 253 - (In the Second draft manuscript) “There are very few studies on the cost of stroke care in Africa. A study in Togo estimated direct cost per person of 936 Euros in only 17 days, about 170 times more than the average annual heath spend of a Togolese. Subsidising and improving post-stroke care may help to reduce stroke case fatality rates and morbidity in Africa”

• Actual Owolabi et al. (2018) article words under subsection Cost of care - “There are very few studies on the cost of stroke care in Africa. A study in Togo estimated direct cost per person of 936 Euros in only 17 days, about 170 times more than the average annual heath spend of a Togolese. Subsidising and improving post-stroke care may help to reduce stroke case fatality rates and morbidity in Africa”. This appears to be a verbatim lift from the source without any paraphrasing

Lines 260-262 (In Second draft manuscript) – “Lifelong assistance, result in an enormous burden, both in human and economic posts. The socioeconomic deprivation is not only associated with stroke and its risk factors, but it is also increasing stroke severity and mortality, and incidence (26).”

• Actual Avan et al. (2017) article words under subsection Background - “Lifelong assistance, resulting in an enormous burden, both in human and economic costs. Evidence suggests that socioeconomic deprivation is not only associated with stroke and its risk factors, but also increases stroke severity [4] and mortality [5], and stroke incidence at younger ages [4].” The phrasing is almost identical.

I would like to kindly suggest that the authors double-check the manuscript for any grammatical errors, referencing issues, and instances of identical wording to original sources. They should either paraphrase properly or use quotation marks for verbatim text where appropriate. Ensuring academic integrity is crucial, therefore, any closely mirrored text should be appropriately paraphrased or enclosed in quotation marks with proper citations, as failure to do so can compromise the credibility of the manuscript.

7. PLOS authors have the option to publish the peer review history of their article (what does this mean? ). If published, this will include your full peer review and any attached files.

**Do you want your identity to be public for this peer review?** For information about this choice, including consent withdrawal, please see our Privacy Policy .

Reviewer #3: **Yes: ** Moyahabo Julius Rampya

---

## [Author Response · Author response to Decision Letter 2]

29 May 2025

Abstract

1. The authors mention that health-related quality of life was assessed at 28 days post-stroke using the EQ-5D-3L questionnaire, but in line 138, it’s stated as 28 days post-discharge. Could you please clarify which timeline is correct?

Response to reviewer comment: Thank you for your observation regarding the timing of the health-related quality of life (HRQoL) assessment. We acknowledge the inconsistency between "28 days post-stroke" and "28 days post-discharge" in the manuscript.

We clarify that the EQ-5D-3L questionnaire was administered 28 days after hospital discharge. This timeline was chosen intentionally to reflect the period in which stroke survivors transition from inpatient care to community and home-based settings a particularly vulnerable phase in low-resource environments such as Mozambique.

Our primary objective was to capture the broader social and economic impact of stroke in daily life, which typically becomes more evident after discharge. In contexts with limited outpatient services and rehabilitation infrastructure, the post-discharge period is critical for understanding quality of life, care gaps, and long-term needs. Therefore, “28 days post-discharge” provides a more accurate and policy-relevant perspective on recovery and support challenges faced by patients and their families.

We have revised the manuscript accordingly to ensure consistency in terminology and avoid confusion.

Introduction

2. Lines 65-66 - The authors state that stroke is now among the top ten causes of death in Mozambique, with a 25% increase between 2009 and 2019. Could you clarify from what position to what position stroke moved in the ranking of causes of death during this period?

Response: Thank you for your question. We appreciate the opportunity to clarify this point.

As stated in the introduction, stroke has become one of the top ten causes of death in Mozambique, with an estimated 25% increase in stroke-related mortality between 2009 and 2019 (IHME, 2020; GBD 2019 Stroke Collaborators, 2021). By 2019, stroke ranked as the second leading cause of death in the country (World Life Expectancy, 2024). However, due to limitations in available historical data, particularly for earlier years such as 2009, we are unable to determine its exact position in the ranking at that time. For this reason, we have chosen to focus on the clear upward trend in stroke-related mortality over the decade, rather than presenting a precise rank transition. This clarification is now explicitly stated in the introduction.

This clarification is now included in the introduction to ensure transparency and accuracy regarding data limitations.

3. Lines 67–81 - the rationale for conducting a cost analysis is articulated, with the focus being on economic burden and resource allocation in Mozambique. However, there is no similar argument presented for assessing quality of life at 28 days post-discharge. Is the quality-of-life assessment intended to contribute to the societal perspective analysis, and if so, could you provide more context on why this was an important outcome to measure in this study?

Materials and Method

Response: Thank you for your thoughtful comment. The assessment of health-related quality of life (HRQoL) at 28 days post-discharge was indeed intended to complement the cost analysis from a societal perspective. Stroke not only imposes a direct economic burden in terms of healthcare costs and productivity losses, but also results in substantial functional impairment that can significantly reduce a person’s ability to live independently and productively.

In the context of Mozambique, where formal rehabilitation services and long-term support systems are limited, understanding post-stroke quality of life is essential to capture the broader societal impact of stroke. By including HRQoL data, we aimed to provide a more comprehensive understanding of patient outcomes and inform policy decisions related to resource allocation, rehabilitation planning, and support services beyond the acute care phase.

We have now clarified this rationale in the manuscript to reinforce the importance of this outcome.

4. Line 86 - Given the small sample size (n=50) and the single-hospital setting, I recommend revising the term ‘representative cohort’ to more accurately reflect the study’s context and limitations. A possible alternative could be: ‘a cohort of first-ever stroke patients admitted to a tertiary-referral hospital…,’ which would appropriately manage the readers’ expectations regarding the study’s generalisability.

Response: Thank you for this valuable suggestion. We agree that the term "representative cohort" may overstate the generalisability of our findings, given the relatively small sample size and the single-hospital setting. We have revised the sentence to reflect a more accurate description of the study population and to better manage readers’ expectations.

5. There seems to be an inconsistency between lines 33 and 100-101 regarding the data collection method. In line 33, it is stated that indirect costs were derived from ‘structured interviews,’ but in the methods section (lines 100-101), it is mentioned that data were actually collected using a ‘semi-structured questionnaire.’ Please consider revising for consistency, these are not the same thing.

Response: Thank you for highlighting this inconsistency. We agree that “structured interviews” and “semi-structured questionnaires” are methodologically distinct. In line with our data collection procedures, we have revised the text for consistency to indicate that indirect cost data were collected using a semi-structured questionnaire administered during interviews.

6. Lines 106 -108 - The authors mention that formal employment is regulated with ‘social protections.’ Could you clarify what types of social protections this includes, especially considering that these may differ in Mozambique compared to other countries?

Response: Thank you for this observation. We acknowledge that the term "social protections" can vary significantly by country. In the Mozambican context, formal employment typically includes access to statutory social security benefits such as retirement pensions, maternity leave, paid sick leave, and work injury compensation, as regulated by the National Institute of Social Security (Instituto Nacional de Segurança Social – INSS). We have revised the manuscript to clarify this point.

7. Line 117 - The authors mention caregiver costs as part of the direct and indirect costs 28 days post-discharge. Could you clarify if the cost of caregivers was standardised, and how you quantified this cost if a family member served as the caregiver? Additionally, was there an assumption made about caregiver costs for patients who couldn’t afford one, or did all participants have formally employed caregivers, or were informal caregiving arrangements (family, friends, etc.) also included?

Response: Thank you for this insightful comment. In our study, caregiver costs were not standardized based on formal wage rates. Most of the caregiving was provided by family members or informal caregivers, and we used a replacement cost approach to estimate the economic value of their time. For these cases, we applied the national minimum wage for the informal sector to approximate the value of unpaid caregiving.

No assumptions were made for patients without available caregivers; in all included cases, some form of caregiving support was reported, whether from family, neighbors, or friends. Therefore, caregiver costs reflected the opportunity cost of time spent providing care rather than actual monetary payments. We have clarified this in the Methods section.

8. Line 139 – typo; a multidimensional concept ‘which’ incorporates…

Response: Thank you for identifying this typo. We have corrected “which” to “that” to maintain grammatical accuracy in the restrictive clause. please see line 165

9. Lines 158 –165 – The authors describe the categorisation of EQ-5D index scores into five levels of HRQoL to facilitate interpretation and statistical modelling. Could the authors please clarify if these categories (< 0 ‘very poor,’ 0-0.5 ‘poor,’ 0.51-0.85‘fair,’ > 0.85 ‘good,’ and 1 ‘perfect state’) were custom defined for this study or if they were based on any previously validated or published classification system in the literature? How did they come up with these?

Response: Thank you for this important observation. The categorization of EQ-5D index scores into five levels of health-related quality of life (HRQoL) — very poor (<0), poor (0–0.5), fair (0.51–0.85), good (>0.85), and perfect (1.0) — was not based on a universally validated or standardized classification. Rather, it was custom-defined for this study to facilitate interpretability and the identification of trends in patient outcomes. This approach was informed by previous literature that similarly grouped EQ-5D scores into categories to support meaningful clinical interpretation, especially in settings where raw index scores are less intuitive for stakeholders (see: McCaffrey et al., 2008; Couser et al., 2021).

We acknowledge that these categories are somewhat arbitrary and may vary between studies. Therefore, we have added a clarification in the Methods section stating that the categories were study-defined for interpretative purposes, and not based on a validated cut-off system.

10. Figure 1 – typo; were there 9 deaths or 10 deaths?

Thank you for pointing this out. We have corrected Figure 1 to reflect this accurately and ensure consistency with the results reported in the text, 10 patients died within the 28-day follow-up period.

Results

11. Table 1 – Misaligned gender and disability values

Response: Thank you for pointing this out. Upon review, we identified a misalignment in the presentation of gender and disability data in Table 1. This occurred due to a formatting error during the manuscript editing process. The values have now been corrected and properly aligned to ensure accuracy and clarity in the presentation of demographic and clinical characteristics. We have updated Table 1 accordingly in the revised version of the manuscript.

12. Line 198 – I was wondering if the costs of care for the comorbidities were separated from the data collected at 28 days post-discharge, or if they were combined into a single cost estimate. Could this potentially introduce confounding effects if all costs were lumped together?

Response: Thank you for this thoughtful question. In our analysis, the costs collected at 28 days post-discharge included all out-of-pocket expenses related to the patient’s health care during that period, including medications, consultations, transport, and caregiving. We did not separate costs specifically attributable to comorbidities, as this information was not consistently reported by participants, and attribution of specific expenses to stroke versus comorbid conditions would have introduced considerable recall bias.

We acknowledge that combining these costs into a single estimate may introduce some confounding, particularly in patients with complex health profiles. However, given the primary objective of estimating the overall post-stroke economic burden from a societal perspective, we opted to report total health-related expenditures. This limitation has now been clarified in the manuscript.

13. Figure 2 - the ‘Estimated Anual’ cost of $US112,225.70 appears to be derived from the per-patient cost for first-time strokes. It’s interesting to see this estimate included without a discussion of its assumptions, such as excluding recurrence and the limitations of the number of stroke cases. Wouldn’t this estimate need to account for the costs of recurrent strokes as well to provide a more accurate reflection of the annual burden? Additionally, does this mean that the estimate would hold if we were to assume that a similar proportion of first-time stroke cases relative to total stroke cases occurs each year?

Response: Thank you for this insightful observation. The estimated annual cost of $US112,225.70 was extrapolated from the per-patient cost of first-ever strokes observed during the study period, and it represents an approximate economic burden based solely on incident cases managed at the study site.

We agree that this estimate does not account for the costs associated with recurrent strokes, nor does it incorporate variability in stroke severity, long-term care, or cases managed at other health facilities. It assumes that the proportion of first-time stroke cases remains stable year-to-year, and that the cost structure and management patterns remain consistent. We acknowledge this as a limitation and have added a clarification in the manuscript to avoid potential misinterpretation of the extrapolated figure.

14. Lines 213-219 and Table 2 - In Line 131 the authors mention that indirect costs included the calculation of days of lost work to assess productivity loss. Since the day-to-day income of informally employed individuals often fluctuates, could you clarify if the income used in the cost assessment was an average calculated across all patients, or if another approach was used to account for income variability?

Response: Thank you for raising this important point. To estimate productivity losses among informally employed participants, we used self-reported average daily income prior to the stroke event. Given the inherent variability in informal income, we collected income data for a typical working day and then triangulated these responses with local minimum wage guidelines and previously published benchmarks from national labour statistics.

We acknowledge that this approach introduces some uncertainty, but it allowed for a more individualized estimate compared to using a uniform wage rate for all informal workers. We have clarified this methodology in the manuscript to improve transparency. Please see line 157.

15. In Table 2 - The authors provide data on monthly income and the total cost of care, but it might be helpful to include as well what percentage of each income group’s monthly income is spent on stroke care costs. This could give a clearer picture of the financial burden on different social classes. Would this analysis be possible to add?

Response: Thank you for this excellent suggestion. We agree that expressing the cost of care as a percentage of patients' monthly income provides a clearer understanding of the financial burden, particularly in a setting where out-of-pocket expenditures can be catastrophic. Based on the available data in Table 2, we have calculated the proportion of monthly income spent on stroke-related costs for each employment category. These values have now been added to the table and discussed in the results section.

Discussion

16. Line 245 – typo; ‘regarding to the author’s knowledge’

Response: Thank you for spotting this typo. We have corrected the phrase “regarding to the author’s knowledge” to “to the best of the authors’ knowledge” to improve grammatical accuracy and clarity. Please see line 316

17. Line 251 – the cost in Togo is reported in Euros, but the rest of the article uses US dollars. For consistency, could this be converted to US dollars?

Response: Thank you for your observation. We agree that presenting all cost estimates in a consistent currency improves clarity and comparability. Accordingly, the cost reported for Togo, originally presented in Euros, has been converted to United States dollars (USD) using the average exchange rate for the year in which the original study was conducted. This adjustment has been made in the revised version of the manuscript, and the source of the exchange rate has been noted in a footnote. Please see line 325

18. Lines 254–262, the authors note that stroke affects younger people in Mozambique and other developing countries (<65 years) but don’t discuss why this might be the case. It would be helpful to explore possible contributing factors, such as a higher burden of infectious diseases (e.g., HIV) etc., rather than just comparing to age statistics from high-income countries without critique.

Response: Thank you for this insightful comment. We agree that the observed younger age of stroke onset in Mozambique and othe

---

## [Decision Letter · Decision Letter 2]

The cost and health-related quality of life of stroke management and care of acutely hospitalized cases in Mozambique

PONE-D-23-21542R2

Dear Dr. Dobe,

We’re pleased to inform you that your manuscript has been judged scientifically suitable for publication and will be formally accepted for publication once it meets all outstanding technical requirements.

Kind regards,

Ryan G Wagner, MSc(Med), MBBCh, PhD

Academic Editor

PLOS ONE

Additional Editor Comments (optional):

Reviewers' comments:

Reviewer's Responses to Questions

**Comments to the Author**

1. If the authors have adequately addressed your comments raised in a previous round of review and you feel that this manuscript is now acceptable for publication, you may indicate that here to bypass the “Comments to the Author” section, enter your conflict of interest statement in the “Confidential to Editor” section, and submit your "Accept" recommendation.

Reviewer #3: All comments have been addressed

2. Is the manuscript technically sound, and do the data support the conclusions?

Reviewer #3: Yes

3. Has the statistical analysis been performed appropriately and rigorously? 

Reviewer #3: I Don't Know

4. Have the authors made all data underlying the findings in their manuscript fully available?

Reviewer #3: Yes

5. Is the manuscript presented in an intelligible fashion and written in standard English?

Reviewer #3: Yes

6. Review Comments to the Author

Reviewer #3: (No Response)

7. PLOS authors have the option to publish the peer review history of their article (what does this mean? ). If published, this will include your full peer review and any attached files.

**Do you want your identity to be public for this peer review?** For information about this choice, including consent withdrawal, please see our Privacy Policy .

Reviewer #3: **Yes: ** Moyahabo J Rampya

---

## [Editor Report · Acceptance letter]

PONE-D-23-21542R2

PLOS ONE

Dear Dr. Dobe,

I'm pleased to inform you that your manuscript has been deemed suitable for publication in PLOS ONE. Congratulations! Your manuscript is now being handed over to our production team.

Kind regards,

on behalf of

Prof. Ryan G Wagner

Academic Editor

PLOS ONE